# PENSIEVE: SELF-SUPERVISED NOVEL VIEW SYNTHESIS VIA IMPLICIT AND EXPLICIT RECONSTRUCTION

## ABSTRACT

Currently almost all state-of-the-art novel view synthesis and reconstruction models rely on calibrated cameras or additional geometric priors for training. These prerequisites significantly limit their applicability to massive uncalibrated data. To alleviate this requirement and unlock the potential for self-supervised training on large-scale uncalibrated videos, we propose a novel two-stage strategy to train a view synthesis model from only raw video frames or multi-view images, without providing camera parameters or other priors. In the first stage, we learn to reconstruct the scene implicitly in a latent space without relying on any explicit 3D representation. Specifically, we predict per-frame latent camera and scene context features, and employ a view synthesis model as a proxy for explicit rendering. This pretraining stage substantially reduces the optimization complexity and encourages the network to learn the underlying 3D consistency in a self-supervised manner. The learned latent camera and implicit scene representation have a large gap compared with the real 3D world. To reduce this gap, we introduce the second stage training by explicitly predicting 3D Gaussian primitives. We additionally apply explicit Gaussian Splatting rendering loss and depth projection loss to align the learned latent representations with physically grounded 3D geometry. In this way, Stage 1 provides a strong initialization and Stage 2 enforces 3D consistency - the two stages are complementary and mutually beneficial. Extensive experiments demonstrate the effectiveness of our approach, achieving high-quality novel view synthesis and accurate camera pose estimation, compared to methods that employ supervision with calibration, pose, or depth information.

## 1 INTRODUCTION

Simultaneously reconstructing the scene and localizing the camera is a long-standing and challenging task in computer vision. Solving this task has the potential to enable the training of fundamental 3D vision networks on large-scale, uncalibrated video data. Previous large reconstruction models typically rely on preprocessed camera parameters and point clouds obtained via SfM (Structure-from-Motion) or SLAM (Simultaneous Localization and Mapping). However, such preprocessing can be time-consuming, require densely sampled views (Schönberger & Frahm, 2016), or depend on additional 3D information (Kerl et al., 2013), which limits their applicability in more challenging and unconstrained datasets.

Recent approaches jointly predict scene and camera parameters with neural networks, training them from scratch on video data. The key challenge lies in differentiably establishing correspondences across views. Earlier depth-based (or point cloud-based) methods (Zhou et al., 2017; Godard et al., 2017) achieve this by employing bilinear interpolation, allowing each projected point to receive gradients from its four neighboring pixels. More recent Gaussian Splatting-based methods (Kang et al., 2024; Hong et al., 2024a) render Gaussian primitives and propagate gradients from pixels within the extent defined by the Gaussian scale. However, these methods still restrict the differentiable matching to a limited spatial neighborhood. Moreover, their carefully designed 3D representations often introduce optimization biases, which lead the network to converge to suboptimal solutions.

In this paper, we propose a two-stage training strategy that combines implicit reconstruction pretraining with explicit reconstruction alignment. Given an input video, our model estimates camera parameters of each frame and predicts context features for a strict subset of those frames. In the pretraining stage, we employ a view synthesis model (inspired by Jin et al. (2025)) to re-render (i.e., predict) all the input frames themselves for self-supervision. This fully end-to-end architecture

avoids the challenges associated with explicit representations and enables implicit scene reconstruction in a self-supervised manner.

However, due to the lack of explicit 3D consistency in pretraining, the learned implicit reconstruction can diverge from the actual physical 3D structure. In essence, the stage1 model behaves like an autoencoder for the target image, with camera parameters acting as an intermediate representation. This means it can only interpolate views at latent cameras, rather than synthesizing views from given real cameras. To address this limitation, we introduce a second training stage to align the implicit reconstruction with the real 3D geometry. This stage additionally predicts explicit 3D Gaussian primitives and computes Gaussian Splatting rendering loss (Kerbl et al., 2023) and depth reprojection loss (Zhou et al., 2017; Godard et al., 2017). This alignment enables both novel view synthesis and accurate camera estimation in an entirely self-supervised setting. Experimental results demonstrate that the two training stages are mutually beneficial: the stage1 pretraining accelerates convergence and improves performance, while the stage 2 alignment forces the network to learn the true 3D structure and camera parameters.

In summary, our contributions are as follows:

• We propose a self-supervised 3D training framework that relies solely on uncalibrated videos, achieving high-quality novel view synthesis (NVS) and accurate camera pose estimation.

• We introduce implicit reconstruction pretraining to address the optimization limitations of prior self-supervised 3D reconstruction methods that rely on explicit representations.

• We propose explicit reconstruction alignment to enforce 3D consistency in the pretrained network, aligning its latent space with the real physical space of the scene.

• We also introduce an interpolated frame enhanced prediction scheme to address the issue of insufficient camera alignment when only two input views are available.

## 2 RELATED WORKS

**Novel View Synthesis.** Novel view synthesis and 3D reconstruction are fundamental computer vision tasks that have been extensively studied, especially since the advent of neural radiance fields (NeRF) (Mildenhall et al., 2021). Traditional approaches such as Structure-from-Motion (SfM) (Agarwal et al., 2011; Schönberger & Frahm, 2016) reconstruct sparse point clouds and estimate camera parameters. Building upon this, new scene representations like NeRF (Mildenhall et al., 2021) and 3D Gaussian Splatting (3DGS) (Kerbl et al., 2023), have been proposed to further improve the quality of NVS. Moreover, many subsequent improvements have been proposed to enhance its rendering quality (Barron et al., 2021; Yu et al., 2024b), geometric accuracy (Li et al., 2023; Huang et al., 2024a), memory efficiency (Yang et al., 2024; Lu et al., 2024), and reconstruction speed (Chen et al., 2022; Müller et al., 2022), as well as to extend it to dynamic scenes (Park et al., 2021; Li et al., 2021b; Fridovich-Keil et al., 2023; Yu et al., 2024a; Huang et al., 2024b).

These representations can be made generalizable by incorporating neural networks. Some methods leverage neural networks to predict generalizable radiance fields (Yu et al., 2021; Chen et al., 2021; Li et al., 2021a; Tian et al., 2023), while others directly predict Gaussian primitives to reconstruct the scene from sparse views in a single feed-forward pass (Charatan et al., 2024; Chen et al., 2024). Beyond architectures based on cost volumes (Chen et al., 2021) or epipolar lines (Suhail et al., 2022; Wang et al., 2021a; Charatan et al., 2024; Chen et al., 2024), several methods adopt fully data-driven approaches, which benefit more from large-scale datasets (Szymanowicz et al., 2024; Tang et al., 2024; Xu et al., 2024; Zhang et al., 2024; Jin et al., 2025). Among them, Jin et al. (2025) propose to directly predict the target frame using a neural network, instead of relying on explicit rendering, thus eliminating the inductive biases introduced by explicit 3D representations. However, these methods require training on datasets with known camera parameters, which limits their applicability to larger-scale, uncalibrated video data.

**Camera-free Novel View Synthesis.** Estimating camera parameters using SfM is not always reliable, especially under sparse-view settings or in scenes with large textureless regions. To address this, several methods have proposed jointly optimizing cameras with NeRF (Wang et al., 2021b; Lin et al., 2021; Meuleman et al., 2023; Jeong et al., 2021; Truong et al., 2023) or 3DGS (Fu et al., 2024; Jiang et al., 2024; Matsuki et al., 2024) during per-scene reconstruction, and leveraging pretrained networks to further improve performance (Bian et al., 2023; Park et al., 2024).

Several methods attempt to train generalizable networks that jointly estimate camera parameters and reconstruct scenes. The difficulty of this setting varies depending on the type of supervision provided during training. Zhang et al. (2025) use ground-truth point clouds and camera parameters for supervision, while NoPoSplat (Ye et al., 2024) and VicaSplat (Li et al., 2025) are trained with given camera parameters. These methods benefit from either direct or indirect camera supervision, thereby alleviating the difficulty of optimization. Smart et al. (2024) leverage the pretrained large-scale reconstruction network (Wang et al., 2024; Leroy et al., 2024) to estimate both point clouds and camera poses, while Hong et al. (2024a) incorporates pretrained depth estimation and matching networks. CoPoNeRF (Hong et al., 2024b) and GGRt (Li et al., 2024) utilize pretrained feature extractors and provide pose to supervised the matching. FlowCam (Smith et al., 2023) employs pretrained optical flow to indirectly supervise both camera estimation and reconstruction. In contrast, SelfSplat (Kang et al., 2024) proposes to jointly optimize a camera network and a 3DGS network from uncalibrated video data without pretrained priors, while provide camera intrinsics to simplify the task. Our method requires only uncalibrated video frames or multi-view images, without relying on any additional data or pretrained priors, achieving high-quality NVS and accurate camera pose estimation, thereby unlocking the potential for training on large-scale and more diverse datasets.

**Concurrent works.** As many recent methods scale up in terms of data, self-supervised approaches that leverage uncalibrated videos have received increasing attention. RayZer (Jiang et al., 2025) provides a detailed discussion of the approach and benefits of implicit reconstruction–based pretraining. Independently, our work introduces a framework with implicit reconstruction and explicit alignment. While demonstrating the strengths of implicit reconstruction, our method further shows that explicit alignment effectively addresses the gap between implicit reconstruction and real physical space. SPFSplat (Huang & Mikolajczyk, 2025) achieves superior performance to supervised methods through self-supervised training, using MASt3R as its initialization. In contrast, our model is trained fully from scratch, highlighting the potential of self-supervised pretraining.

## 3 METHOD

### 3.1 DEFINITION

Given an uncalibrated video with a length of $N$ frames $\{\mathbf{I}_i \mid i = 1, ..., N\}$, our network $\mathcal{M}_\theta$ predicts, for each frame $\mathbf{I}_i$, the corresponding context features $\mathbf{F}_i^c$, pixel-aligned Gaussian primitives $\mathbf{G}_i$, and camera intrinsic $\mathbf{K}_i$ and extrinsic $\mathbf{P}_i$. Mathematically, this can be formulated as follows:

$$\{\mathbf{F}_i^c, \mathbf{G}_i, \mathbf{K}_i, \mathbf{P}_i | i = 1, ..., N\} = \mathcal{M}_\theta(\{\mathbf{I}_i | i = 1, ..., N\}),\tag{1}$$

where the prediction for each frame is conditioned on all frames, allowing the model to incorporate multi-view context in its estimation.

Thus, given a target camera $\{\mathbf{K}_t, \mathbf{P}_t\}$, we enable a unified model $\mathcal{R}^M$ to synthesize the corresponding target view image $\hat{\mathbf{I}}_t^M$, similar to Jin et al. (2025). In addition, we can leverage Gaussian Splatting Rasterization $\mathcal{R}^G$ (Kerbl et al., 2023) to render the target view image $\hat{\mathbf{I}}_t^G$:

$$\hat{\mathbf{I}}_t^M = \mathcal{R}^M(\mathbf{K}_t, \mathbf{P}_t, \mathbf{F}_{1:N}^c, \mathbf{K}_{1:N}, \mathbf{P}_{1:N}) \qquad \hat{\mathbf{I}}_t^G = \mathcal{R}^G(\mathbf{K}_t, \mathbf{P}_t, \mathbf{G}_{1:N})\tag{2}$$

For the definition of the camera, we assume an ideal pinhole camera model with the principal point at the image center. The network predicts the unknown focal lengths $f_x$ and $f_y$. For the extrinsics $\mathbf{P}$, the network directly predicts the rotation as a quaternion $\mathbf{R}^q \in \mathbb{R}^4$, along with the translation $\mathbf{t} \in \mathbb{R}^3$. We assume that all frames in a given video share the same intrinsic parameters, which is reasonable for most video sequences. To achieve this, we simply average the predicted intrinsic parameters across all frames.

For each Gaussian primitive $\mathbf{G} = \{\boldsymbol{\mu}, \alpha, \mathbf{q}, \mathbf{s}, \mathbf{c}\}$, we adopt the geometrically accurate 2D Gaussian Splatting formulation proposed in Huang et al. (2024a). Consequently, our network is required to predict the following parameters for each primitive: the center position $\boldsymbol{\mu} \in \mathbb{R}^3$, the opacity $\alpha \in \mathbb{R}$, the rotation represented as a quaternion $\mathbf{q} \in \mathbb{R}^4$, the anisotropic scale $\mathbf{s} \in \mathbb{R}^2$, and the color modeled via spherical harmonics (SH) coefficients $\mathbf{c} \in \mathbb{R}^k$, where $k$ denotes the number of SH coefficients used to represent view-dependent appearance.

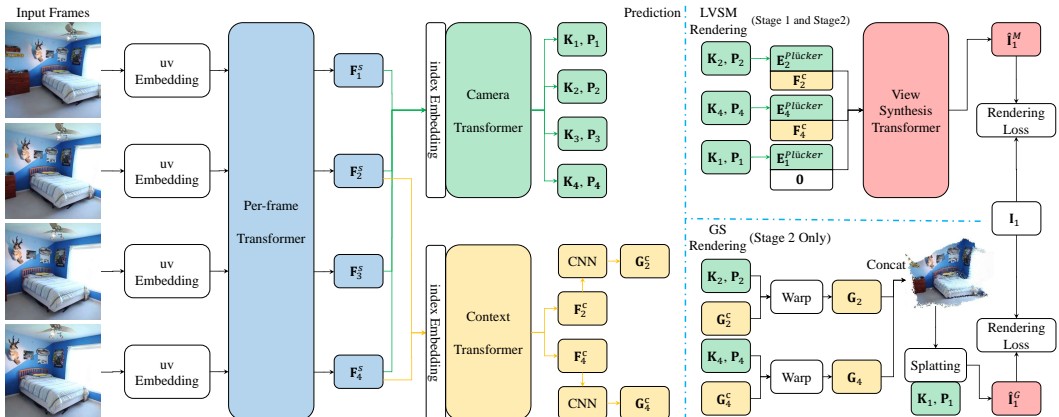

Figure 1: **Pipeline**. Given an $N$-frame uncalibrated video (with $N = 4$ shown), our model predicts per-frame camera parameters $(\mathbf{K}, \mathbf{P})$, and for a proper subset of context views (e.g., 2, 4), predicts context features $\mathbf{F}^c$ and pixel-aligned Gaussian primitives $\mathbf{G}^c$. These are used for LVSM implicit rendering and GS explicit rendering, respectively. We iteratively rerender all frames and compute losses against the input video.

## 3.2 NETWORK ARCHITECTURE

Although our model $\mathcal{M}_\theta$ is designed to output per-frame predictions as formulated in Eq. (1), it is important to note that during self-supervised training, the context frames must form a proper subset of the frames used for camera prediction, as we will discuss in Sec. 3.3. Therefore, our model should be structured into two components: the Camera Transformer, which predicts the camera parameters, and the Context Transformer, which reconstructs the scene (implicitly and explicitly), with the constraint that the input to the Context Transformer is always a strict subset of the frames fed to the Camera Transformer during training. Naturally, since both components require processing image features, we design a shared Per-Frame Transformer to extract per-frame image features in advance, reducing the overall parameters. We also implement the LVSM module in Eq. (2) using the same architecture. As shown in Fig. 1, all four transformers in our pipeline share the same architecture type, similar to GS-LRM (Zhang et al., 2024), but differ in embedding methods, number of layers, and hidden dimensions.

**Transformer.** All of our transformers transform the input maps into output maps of the same resolution. In each transformer, we first patchify the input maps and flatten each patch into tokens. Following GS-LRM (Zhang et al., 2024), each transformer is composed of stacked consecutive blocks, where each block can be described as:

$$\mathbf{T}_l^m = \text{SelfAttn}(\text{LN}(\mathbf{T}_{l-1})) + \mathbf{T}_{l-1} \quad \mathbf{T}_l = \text{MLP}(\text{LN}(\mathbf{T}_l^m)) + \mathbf{T}_l^m, \tag{3}$$

where $\mathbf{T}$ denotes the tokens at each layer, $l$ indicates the current layer index, SelfAttn refers to the self-attention module, MLP is a multi-layer perceptron, and LN stands for layer normalization. After passing through $L$ transformer blocks, we unpatchify $\mathbf{T}_L$ to reconstruct output maps of the same resolution as the input. We now describe the differences among the four Transformers.

For the **Per-Frame Transformer**, we feed each frame independently into the network to avoid any leakage of inter-frame information. We simply use the pixel coordinates as the positional embedding. Specifically, each input map is augmented with two additional channels whose values are the normalized pixel coordinates $(u/W, v/H)$, where $(u, v)$ denotes the coordinates, and $(W, H)$ represents the resolution of the map.

For the **Camera Transformer**, we treat all image features as a group of input maps. For each map in this group, we simply use its index as the positional embedding. Specifically, each input map is augmented with one additional channel whose value is $i/N$, where $i$ denotes the index of the current map within the group, and $N$ is the total number of maps. We simply apply Global Average Pooling (GAP) over each output map, followed by a lightweight MLP to predict the camera parameters.

For the **Context Transformer**, we randomly sample a strict subset of the image features as input. We similarly apply index embedding to the input. The predicted context features $\mathbf{F}^c$ are directly fed into a lightweight CNN to predict pixel-aligned Gaussians $\mathbf{G}^c$ in the camera coordinate system.

For the **View Synthesis Transformer**, we follow the approach of Jin et al. (2025) by representing all cameras using Plücker coordinates embedding. We apply the Plücker coordinates embedding of the corresponding camera to the context features as an implicit reconstruction of the scene. For the target camera, we concatenate its embedding with a zero map $\mathbf{0}$ to form the target tokens. Finally, we take only the updated target tokens from the output as the synthesized target view image.

### 3.3 LATENT RECONSTRUCTION PRETRAINING

Training networks to estimate camera parameters and scene structure solely from uncalibrated videos or multi-view images has long been a fundamental challenge in computer vision. The key is differentiably identifying correspondences between images. Early depth-based (Zhou et al., 2017; Godard et al., 2019) and plane-based (GonzalezBello & Kim, 2020; Wang et al., 2023) methods employed bilinear interpolation to make the matching differentiable:

$$\hat{\mathbf{I}}_t^{u,v} = \mathbf{I}_c[\pi(\mathbf{K}_c, \mathbf{P}_c, \pi^{-1}(\mathbf{K}_t, \mathbf{P}_t, D_t^{u,v}))], \tag{4}$$

where $D$ denotes the depth, $\mathbf{K}, \mathbf{P}$ represent the camera parameters, $\pi(\cdot)$ is the projection function, and $\mathbf{I}_c[\cdot]$ indicates bilinear sampling on the context image. After a target pixel is projected into the context view, its gradients can only be computed from the four neighboring pixels through bilinear interpolation, which significantly limits its ability to establish accurate correspondences.

Gaussian Splatting-based approaches (Kang et al., 2024) have leveraged rendered Gaussian primitives to establish correspondences, yet each point is still only affected by the pixels within its Gaussian scale range. These limitations increase the optimization difficulty and often trap the network in suboptimal solutions.

Jin et al. (2025) address this by introducing an end-to-end network that directly renders (predicts) the target image from the target camera and the context images, allowing each target pixel to be differentiable with respect to all context pixels and thereby circumventing the locality limitations of explicit 3D representations. Motivated by this, we adopt a view synthesis model $\mathcal{R}^M$ as the renderer for network pretraining, enabling the network to implicitly learn better correspondences. Note that this model is also trained from scratch.

Given the input image set, we partition it into a context set and a target set, $\mathcal{I} = \mathcal{I}^c \cup \mathcal{I}^t$. The context set $\mathcal{I}^c$ is used to extract the context features $\mathcal{F}^c$, while both sets are jointly used to estimate the camera set $\mathcal{C}$, ensuring that all cameras lie in a consistent coordinate space.

$$\mathcal{F}^c = \mathcal{M}_\theta^{\text{ctx}}(\mathcal{I}^c) \quad \mathcal{C} = \{\mathcal{C}^c, \mathcal{C}^t\} = \mathcal{M}_\theta^{\text{cam}}(\mathcal{I}^c \cup \mathcal{I}^t), \tag{5}$$

where $\mathcal{M}_\theta^{\text{ctx}}$ and $\mathcal{M}_\theta^{\text{cam}}$ denote the context branch and the camera branch of our network $\mathcal{M}_\theta$, respectively. Then, by selecting any camera $(K_i, P_i) = C_i \in \mathcal{C}$, we can reconstruct the corresponding input image:

$$\hat{\mathbf{I}}_i^M = \mathcal{R}^M(C_i, \mathcal{F}^c, \mathcal{C}^c). \tag{6}$$

We reconstruct all input images by iterating over $i = 1, \ldots, N$, and compute the loss with the original inputs $\mathcal{I}$.

Unlike explicit 3D reconstruction, the view-synthesis model $\mathcal{R}^M$ does not project any explicit 3D geometry. Instead, it directly predicts the target image, making the entire process closer to an encoding–decoding pipeline, with the latent camera parameters serving as an intermediate representation. Each image in the target set is first compressed into the latent camera parameters, which is then decoded back to the original image by querying from the context images, even though this latent camera representation does not necessarily align with the real physical camera. We convert the camera parameters into the corresponding Plücker embedding before feeding them into the network.

In general, once a scene is reconstructed, specifying a camera uniquely determines the rendered image. Therefore, the degrees of freedom (DoF) of the camera parameters correspond to the minimal DoF required to encode a target image. Fewer DoF would lead to insufficient information for querying, while excessive DoF may cause information leakage, allowing the network to bypass scene reconstruction and instead learn a shortcut mapping from inputs to outputs. Our pretraining design follows this principle by directly using the intermediate representation as the camera.

However, there is one critical exception that can break this encoding mechanism—when the target image itself is included among the context images:

$$\hat{\mathbf{I}}_i^M = \mathcal{R}^M(C_i, \mathcal{M}_\theta^{\text{ctx}}(\mathcal{I}^c), \mathcal{C}^c) \quad \mathbf{I}_i \in \mathcal{I}^c, \tag{7}$$

Since the context branch $\mathcal{M}_\theta^{\text{ctx}}$ does not perform any compression, the channel dimension of the resulting context features can even exceed that of the original context images, the network can trivially encode the 1D index of the target image $\mathbf{I}_i$ within the context set $\mathcal{I}^c$ into $C_i$, simply copying the input to the output and bypassing any meaningful reconstruction. To prevent this, we ensure throughout our training process that the context images are always a strict subset of the input video frames, so that at least one target image lies outside the context set.

Figure 1 illustrates the pipeline of our approach. Each input frame is first encoded into its corresponding camera parameters, and we randomly sample a subset of frames as the context set to extract their features, as described in Eq. (5). Then, as shown in Eq. (6), we decode all input frames sequentially by iterating over $i = 1, \ldots, N$. We compute the rendering loss as a combination of an MSE and an LPIPS loss:

$$\mathcal{L}_{\text{render}}^M = \frac{1}{\sum w} \sum_{i=1}^N w_i (\text{MSE}(\hat{\mathbf{I}}_i^M, \mathbf{I}_i) + \lambda_1 \, \text{LPIPS}(\hat{\mathbf{I}}_i^M, \mathbf{I}_i)) \quad w_i = \begin{cases} 1, & \mathbf{I}_i \notin \mathcal{I}^c \\ w_{\text{low}}, & \mathbf{I}_i \in \mathcal{I}^c \end{cases}, \quad (8)$$

where we assign a lower weight (e.g., $w_{\text{low}} = 0.1$) to the loss of reconstructed context images to mitigate the trivial encoding issue.

### 3.4 EXPLICIT RECONSTRUCTION ALIGNMENT

Although the implicit reconstruction pretraining avoids the optimization challenges associated with explicit representations, it inherently lacks explicit 3D consistency constraints. As a result, the reconstructed space may diverge from the true physical world. While the cameras used during pretraining could be physically correct, it is more likely that the network leverages the same DoF to encode unintelligible latent features. Therefore, additional explicit 3D alignment is required to enforce 3D consistency.

Similar to SelfSplat (Kang et al., 2024), we choose to use the Gaussian Splatting rendering loss together with a depth projection loss to enable self-supervised alignment. We use the latent pretrained weight and additionally predict pixel-aligned Gaussian primitives from the context features using a lightweight CNN, as illustrated in Fig. 1. The center position $\boldsymbol{\mu}$ of each Gaussian primitive is obtained by first predicting a depth map and then back-projecting it into 3D space, formulated as:

$$\boldsymbol{\mu}_i^{u,v} = \mathbf{R}_i D_i^{u,v} \mathbf{K}_i^{-1} [u \, v \, 1]^\top + \mathbf{t}_i, \quad (9)$$

where $D_i^{u,v}$ is the predicted depth at pixel $(u, v)$, $\mathbf{K}_i$ and $\mathbf{R}_i, \mathbf{t}_i$ are the intrinsic and extrinsic camera parameters for frame $i$, respectively. Importantly, the depth map $D_i$ is also predicted by the CNN trained from scratch.

In the stage 2 training, we retain the pretraining loss Eq. (8) while additionally introducing the Gaussian Splatting rendering loss $\mathcal{L}_{\text{render}}^G$. Specifically, we concatenate all Gaussian primitives predicted from the context features and rerender the entire input video $\{\hat{\mathbf{I}}_i^G \mid i = 1, \ldots, N\}$. The Gaussian Splatting rendering loss $\mathcal{L}_{\text{render}}^G$ follows the same formulation as Eq. (8), except that $\hat{\mathbf{I}}_i^M$ is replaced by $\hat{\mathbf{I}}_i^G$.

For predicted depth $\mathbf{D}$ by our model and the rendered depth $\hat{\mathbf{D}}$ by the Gaussian Splatting, we compute the projection loss (Zhou et al., 2017) and the edge-aware smoothness loss (Godard et al., 2017). Given the depth map for the $i$-th frame, we use the camera parameters to compute the per-pixel projection onto the $j$-th frame. We then synthesize the $i$-th frame $\hat{\mathbf{I}}_i^D$ by bilinearly sampling colors from the $j$-th frame and compute the MSE loss between the synthesized and ground-truth images. The projection and smoothness losses are:

$$\mathcal{L}_{\text{proj}} = \text{MSE}(\hat{\mathbf{I}}_i^D, \mathbf{I}_i) \quad \mathcal{L}_{\text{ds}} = |\partial_x \mathbf{D}_i \odot e^{-\gamma \|\partial_x \mathbf{I}_i\|_1}| + |\partial_y \mathbf{D}_i \odot e^{-\gamma \|\partial_y \mathbf{I}_i\|_1}|, \quad (10)$$

where $\gamma$ controls the smoothness around edges (GonzalezBello & Kim, 2020). We apply the same loss to the rendered depth $\hat{\mathbf{D}}$, simply replacing $\mathbf{D}$ with $\hat{\mathbf{D}}$ in the Eq. (10). It is important to note that each input frame has an associated rendered depth map $\hat{\mathbf{D}}$, whereas only the context frames have corresponding predicted depth maps $\mathbf{D}$. For each projection, the frame $j$ is randomly sampled from the set of input frames.

Therefore, the final loss for our stage 2 training is:

$$\mathcal{L}_{\text{stage2}} = \mathcal{L}_{\text{render}}^{M} + \mathcal{L}_{\text{render}}^{G} + \lambda_2 \mathcal{L}_{\text{proj}} + \lambda_3 \mathcal{L}_{\text{ds}}, \tag{11}$$

which is averaged over pixels, views, and batches. $\lambda_1$ and $\lambda_2$ are hyperparameters used to balance different loss terms. Notably, $\lambda_2$ is gradually reduced to zero during training, as the projection loss is affected by the occlusion problem (Godard et al., 2019).

## 3.5 Interpolated Frame Enhanced Prediction

During training, we randomly sample the length of the input video, with a minimum of two frames, and shuffle the input frame order to simulate the multi-view reconstruction task. However, when only two frames are provided as input, the context frame must be a strict subset of the input frames, leaving only one frame available as the context. In this case, there are large unseen regions when rendering the other frame. Gaussian Splatting produces large holes when rendering these unseen regions, causing the network to predict oversized Gaussian primitives and underestimate camera motion in an attempt to fill the holes. As a result, alignment performance degrades under two-frame input conditions.

To address this issue, after completing the two-stage training, we apply a specific inference-time strategy tailored for the two-frame case. At inference, we interpolate the two input frames into a three-frame video and re-feed it into the network to obtain the final output. This interpolation is performed using our LVSM rendering. Specifically, after the initial two-frame prediction, we average the predicted camera parameters of the two input frames to generate an intermediate camera, which serves as the target camera. Using LVSM rendering in Eq. (2), we render the interpolated middle frame, which is then combined with the original two frames to form a three-frame video that is re-input into the network for standard prediction.

## 4 Experiments

### 4.1 Datasets and Implementation Details

Our model is trained solely on raw video frames, disregarding any auxiliary preprocessed data. During training, input frame orders are shuffled. Please refer to the Sec. C for more details.

**RealEstate10K.** The RealEstate10K (Zhou et al., 2018) dataset contains 80,000 clips from 10,000 YouTube videos. We follow the train/test split used in PixelSplat (Charatan et al., 2024) and train at a resolution of $256 \times 256$, using video clips of random lengths $N \in [2, 7]$.

**DL3DV-10K.** The DL3DV-10K (Ling et al., 2024) dataset contains 10,510 videos. Following the protocol of Kang et al. (2024), we finetune all the models on DL3DV using the weights trained on RealEstate10K (Zhou et al., 2018). Evaluation is performed on the DL3DV-140 benchmark, using the small-overlap and large-overlap splits from Hong et al. (2024a). Finetuning is performed at a resolution of $256 \times 448$, with random clip lengths $N \in [2, 5]$.

### 4.2 Evaluation Protocols

For all comparison methods, we follow their original settings and provide the additional data required by each method for training and evaluation to ensure fairness. Our evaluation protocols also follow their settings, using two frames as context frames and selected intermediate frames as test frames (specified in Charatan et al. (2024); Hong et al. (2024a)). Based on how the camera pose of the test view is obtained, we categorize the evaluation into Target-aware Evaluation (Hong et al., 2024a; Kang et al., 2024) and Target-aligned Evaluation (Hong et al., 2024b). As a contrast, our method is trained on variable-length, unordered video frames, without any fine-tuning tailored to such inputs or additional data.

**Target-aware Evaluation**. Following Hong et al. (2024a); Kang et al. (2024), we predict the camera of the test view using both the context images and the test image as inputs to the trained camera pose network. This setting provides greater overlap between views for pose estimation. However, as discussed in Sec. 3.3, this acutally encodes and decodes the target view, rather than performing true novel view synthesis.

Table 1: Quantitative Comparison on RealEstate10K (Zhou et al., 2018). The best is in **bold** and the second is underlined in each metric. **K**, **P**, **D**, **M** and **F** denote additional training data of camera intrinsics (**K**), camera extrinsics (**P**), depth (**D**), matching (**M**) and optical flow (**F**), respectively. * stands for our method with optimized camera pose of the test view.

| Method | Training Data | PSNR↑ | SSIM↑ | LPIPS↓ | RRA@5↑ | RRA@15↑ | RTA@5↑ | RTA@15↑ |
|---|---|---|---|---|---|---|---|---|
| RealEstate10K Zhou et al. (2018) Target-aware Evaluation | | | | | | | | |
| FlowCam | Video + **F** | 20.63 | 0.678 | 0.343 | 47.2 | 87.9 | 2.99 | 21.4 |
| PF3plat | Video + **KDM** | 22.84 | 0.790 | 0.190 | **92.8** | **98.3** | **58.6** | **91.5** |
| SelfSplat | Video + **K** | 22.04 | 0.772 | 0.237 | 70.9 | 92.1 | 37.1 | 67.0 |
| Ours-explicit | Video | 23.21 | 0.784 | 0.170 | 84.8 | 98.2 | 48.1 | 85.2 |
| Ours | Video | **26.53** | **0.843** | **0.115** | 84.8 | 98.2 | 48.1 | 85.2 |
| RealEstate10K Zhou et al. (2018) Target-aligned Evaluation | | | | | | | | |
| CoPoNeRF | Video + **KP** | 19.66 | 0.668 | 0.327 | 89.8 | **98.4** | 47.0 | **85.8** |
| PF3plat | Video + **KDM** | 20.04 | 0.643 | 0.260 | **92.0** | 97.7 | 37.8 | 85.6 |
| SelfSplat | Video + **K** | 18.50 | 0.598 | 0.347 | 60.1 | 85.1 | 24.1 | 49.4 |
| Ours-explicit | Video | 21.36 | 0.693 | 0.214 | 85.5 | 97.7 | **50.3** | 85.3 |
| Ours-explicit* | Video | 23.11 | 0.763 | 0.182 | 85.5 | 97.7 | **50.3** | 85.3 |
| Ours | Video | 22.20 | 0.712 | 0.176 | 85.5 | 97.7 | **50.3** | 85.3 |
| Ours* | Video | **23.96** | **0.778** | **0.145** | 85.5 | 97.7 | **50.3** | 85.3 |

Table 2: Quantitative Comparison on DL3DV-140 Ling et al. (2024). The best is in **bold** in each metric. *: stands for our method with optimized camera pose of the test view.

| Method | small | | | | | large | | | | |
|---|---|---|---|---|---|---|---|---|---|---|
| | PSNR↑ | SSIM↑ | LPIPS↓ | RRA@5↑ | RTA@5↑ | PSNR↑ | SSIM↑ | LPIPS↓ | RRA@5↑ | RTA@5↑ |
| DL3DV-Benchmark Ling et al. (2024) Target-aware Evaluation | | | | | | | | | | |
| PF3plat | 18.77 | 0.583 | 0.291 | 71.9 | 39.6 | 21.35 | 0.679 | 0.223 | 89.9 | 46.0 |
| SelfSplat | 19.11 | 0.586 | 0.396 | 74.1 | 46.0 | 21.70 | 0.689 | 0.309 | 89.9 | 58.9 |
| Ours | **21.77** | **0.662** | **0.183** | **84.9** | **58.3** | **23.89** | **0.750** | **0.133** | **93.5** | **59.0** |
| DL3DV-Benchmark Ling et al. (2024) Target-aligned Evaluation | | | | | | | | | | |
| PF3plat | 17.01 | 0.466 | 0.345 | 76.3 | 33.1 | 19.80 | 0.591 | 0.257 | 90.7 | 44.6 |
| SelfSplat | 16.99 | 0.478 | 0.451 | 61.9 | 36.7 | 20.32 | 0.617 | 0.340 | 88.5 | 48.9 |
| Ours | 19.36 | 0.543 | 0.242 | **82.7** | **55.4** | 22.21 | 0.673 | 0.164 | **92.8** | **61.9** |
| Ours* | **20.02** | **0.587** | **0.222** | **82.7** | **55.4** | **22.52** | **0.696** | **0.153** | **92.8** | **61.9** |

**Target-aligned Evaluation**. Following Hong et al. (2024b), only context views are given as input to the camera network to estimate pose. The ground-truth camera parameters of the test view are then aligned to the estimated context camera poses for rendering. This ensures a more realistic evaluation of the NVS capabilities. Since our input views are sparse and no prior information is introduced, the reconstruction may not align with the ground truth, leading to discrepancies in the aligned test-view poses. Therefore, following Lin et al. (2021); Wang et al. (2021b), we freeze all parameters except the test-view extrinsics during evaluation and apply the Gaussian Splatting rendering loss for 40 optimization iterations to further refine the alignment. We present the results with both optimized and non-optimized camera poses for comparison.

We evaluate the novel view synthesis performance using PSNR, SSIM, and LPIPS metrics, and use the Relative Rotation Angle (RRA) and Relative Translation Angle (RTA) to evaluate the predicted poses between pairs of input views. The @5 and @15 metrics indicate the percentage of view pairs with angular errors within 5° and 15°, respectively. In the target-aware evaluation, SelfSplat and PF3plat support only a single target view at a time, forming an input triplet $\{\mathbf{I}_0^c, \mathbf{I}^t, \mathbf{I}_1^c\}$. In contrast, the commonly used test split in prior reconstruction works (Charatan et al., 2024; Chen et al., 2024; Zhang et al., 2024) provides three target views for each pair of context views. Therefore, for each context pair, our target-aware evaluation is performed three times, once for each of the three target views.

### 4.3 PERFORMANCE COMPARISONS

By default, all our NVS evaluations are rendered using the implicit branch. The explicit 3D Gaussian branch is only employed to align the latent space with the physical world. To this end, we adopt the 2DGS representation and assign a higher weight to the depth smoothness loss, focusing more on the geometry of the explicit branch rather than its rendering. Nevertheless, for completeness, we also report the rendering results of the explicit branch $\hat{\mathbf{I}}^G$ in Tab. 1 as a reference.

The quantitative results on RealEstate10K (Zhou et al., 2018) shown in Tab. 1 demonstrate that, although our method is trained only on raw video frames, it achieves the best novel view synthesis

| Context View $\mathbf{I}_1$ | Context View $\mathbf{I}_2$ | GT | CoPoNeRF | PF3plat | SelfSplat | Ours |

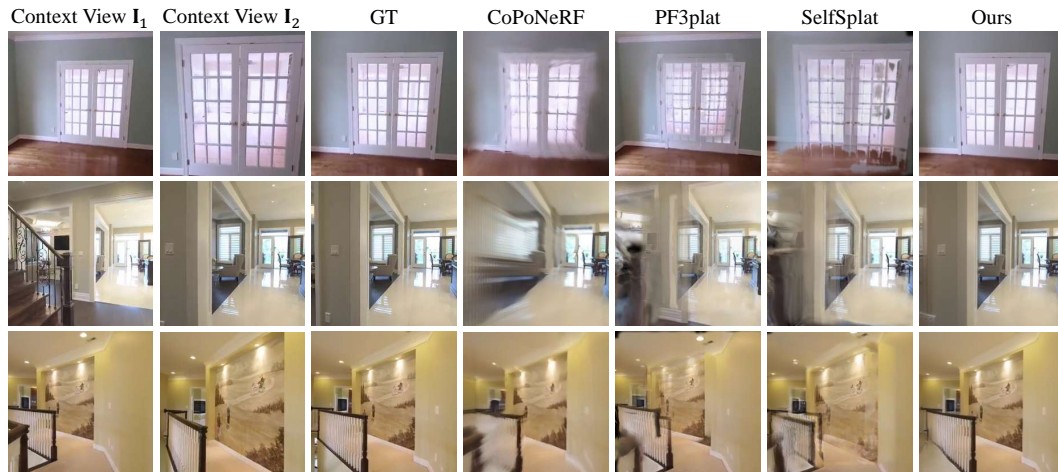

Figure 2: Novel view synthesis comparisons on RealEstate10K Zhou et al. (2018). Our method better aligns with ground-truth poses and image content.

quality, with slightly lower camera pose estimation accuracy. This is because CoPoNeRF (Hong et al., 2024b) benefits from additionally provided camera extrinsics as supervision for pose estimation, while PF3plat (Hong et al., 2024a) leverages a pretrained matching network (Lindenberger et al., 2023) that supplies rich correspondence information. The qualitative results in Fig. 2 further show that our method accurately aligns the target pose and synthesizes the image, even from two input frames with large textureless regions.

The quantitative results on DL3DV (Ling et al., 2024) shown in Tab. 2 indicate that our method achieves the best performance in both novel view synthesis and camera pose estimation. This is attributed to the increased difficulty of the DL3DV dataset, where PF3plat (Hong et al., 2024a) is constrained by the accuracy of its pretrained matching network (Lindenberger et al., 2023), which may fail for pose estimation, and SelfSplat (Kang et al., 2024) suffers from the optimization limitations of explicit Gaussian primitives. In contrast, our method benefits from implicit reconstruction to learn latent correspondences through a fully end-to-end network, leading to superior performance. Even without any camera information as guidance, by introducing the Stage 2, our method still achieves the best performance for camera pose estimation.

The existing self-supervised frameworks used as baselines primarily target the two-context-view setting and focus only on rendering quality. To demonstrate the effectiveness of our model on variable-length videos, we conduct experiments on the RealEstate10K (Zhou et al., 2018) dataset using six-frame inputs. Furthermore, to evaluate the generalization ability of our approach and the accuracy of the learned depth, we perform zero-shot depth estimation on the KITTI (Geiger et al., 2012) dataset. Please refer to Sec. B.1 for details.

### 4.4 ABLATION STUDIES

We conduct ablation studies on the RealEstate10K (Zhou et al., 2018) dataset. Since different components are best evaluated under different protocols, we report results under the Target-aligned Evaluation in Tab. 3, and results under the Target-aware Evaluation in Tab. 4. It is worth noting that we initialize camera rotations as identity matrices. Given that the RealEstate10K (Zhou et al., 2018) dataset contains a large number of scenes with minimal camera rotation, the RRA@5 metric yields a seemingly high value of 36.6% for both the Untrained model and models that fail to converge.

In Tab. 3, the result of w/o Stage 1 shows that our method fails to converge without the implicit reconstruction pretraining. Our proposed implicit reconstruction stage avoids the optimization difficulties of explicit representations and enables the model to learn underlying matching, thereby facilitating convergence. The result of w/o Stage 2 indicates that the implicitly reconstructed space is not aligned with the real physical space, leading to failures in performing novel view synthesis under a given pose. IF refers to the Interpolated Frame Enhanced Prediction described in Sec. 3.5. By naturally in-

Table 3: Target-aligned Ablation Studies.

| Method | PSNR↑ | SSIM↑ | LPIPS↓ | RRA@5↑ | RTA@5↑ |
|---|---|---|---|---|---|
| w/o Stage 1 | 12.97 | 0.401 | 0.529 | 36.6 | 2.28 |
| w/o Stage 2 | 13.15 | 0.490 | 0.378 | 62.0 | 0.10 |
| w/o IF | 23.25 | 0.752 | 0.163 | 78.5 | 41.0 |
| Full | **23.96** | **0.778** | **0.145** | **85.5** | **50.3** |

Table 4: Target-aware Ablation Studies.

| Method | PSNR↑ | SSIM↑ | LPIPS↓ | RRA@5↑ | RTA@5↑ |
|---|---|---|---|---|---|
| Untrained | 10.66 | 0.037 | 0.830 | 36.6 | 0.00 |
| w/o Stage 1 | 15.71 | 0.479 | 0.448 | 36.6 | 2.36 |
| w/o Stage 2 | **27.30** | **0.858** | **0.107** | 64.2 | 0.10 |
| Full | 26.62 | 0.846 | 0.113 | **88.2** | **53.1** |

terpolating frames through interpolated camera poses, our method mitigates the under-convergence issue of the camera network when using only two input views, thus improving performance. Fig. 3 shows that without Stage 1, the model fails to learn meaningful Gaussian primitives, demonstrating the effectiveness of Stage 1 implicit reconstruction.

Table 4 presents the impact of different training strategies. Consistent with earlier observations, the w/o Stage 1 result again highlights that our stage 1 pretraining facilitates network convergence. The w/o Stage 2 result demonstrates that our network is capable of effectively implicitly reconstructing the scene and using the latent camera representation to predict the target image. Interestingly, due to our use of a physically meaningful Plücker embedding as the camera representation, the predicted latent camera rotations closely approximate the ground-truth rotations, leading to an improved RRA@5 score compared with the untrained model. Qualitatively, we also observe that the latent camera translations tend to align with the ground-truth translations, as shown in Fig. 4. After applying explicit reconstruction alignment, the Full setting demonstrates accurate camera pose estimation, highlighting the effectiveness of our self-supervised alignment strategy. Meanwhile, the image synthesis quality shows a slight degradation due to enforced 3D consistency.

| Target Image | w/o stage1 | w/ stage1 | First Frame $\mathbf{I}_1$ | Last Frame $\mathbf{I}_7$ | w/o stage2 | w/ stage2 |

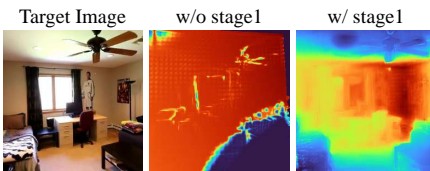

Figure 3: Comparison of the rendered depth maps with and without Stage 1 pretraining.

Figure 4: Comparison of the camera trajectories with and without Stage 2 alignment.

## 5 CONCLUSION

We propose a two-stage training strategy for learning novel view synthesis models from uncalibrated videos. In the first stage, we perform implicit reconstruction pretraining, which addresses the optimization limitations of explicit 3D representations and enables the network to implicitly learn better correspondences. However, since the latent space is not necessarily aligned with the real 3D world, it limits the model to performing novel view synthesis from any given camera. To address this, we introduce explicit reconstruction alignment in the second stage. Specifically, we predict explicit Gaussian primitives and additionally compute the 3DGS rendering loss and depth reprojection loss, injecting 3D consistency in a fully self-supervised manner. To further alleviate the issue of insufficient camera alignment when using only two input frames, we propose an interpolation-based prediction strategy. Experimental results demonstrate that our pretraining phase facilitates network convergence, while the second stage effectively aligns the latent reconstruction with the real 3D space, resulting in high-quality novel view synthesis and accurate camera pose estimation.

## 6 STATEMENT

**Ethics Statement.** Our approach does not involve generative capability and is evaluated solely on publicly available scene datasets. When extending the method to generative tasks or larger datasets, it is important to ensure data integrity and content safety. When deployed in high-precision systems, the reconstruction quality should be carefully evaluated and further improved if necessary.

**Reproducibility Statement.** We provide detailed descriptions of the data, implementation, and training details in Sec. 4.1 and Sec. C. The source code is included in the supplementary material to facilitate reproducibility.

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

## A  INTRODUCTION OF APPENDIX

We provide additional experiments and implementation details, along with discussions of limitations and the use of LLMs. The supplementary material also contains videos for clearer visualization and code to support reproducibility.

## B  ADDITIONAL EXPERIMENTS

### B.1  ADDITIONAL PERFORMANCE COMPARISON

To evaluate the performance of our model with varying numbers of input frames, we conduct experiments on the RealEstate10K (Zhou et al., 2018) dataset using six-frame inputs, with results shown in Tab. 5. For each test video, we sample six context views at intervals of 48 frames. For videos too short to meet this interval, we adopt the maximum possible interval. For novel view synthesis, we compare against the MVS-based method MVSplat (Chen et al., 2024). For pose estimation, we compare against DUSt3R (Wang et al., 2024), which is trained with supervision on large-scale datasets. Since both our method and DUSt3R can estimate intrinsic focal parameters, we report the absolute relative error (AbsRel) for horizontal focal length $f_x$ and vertical focal length $f_y$. The results show that DUSt3R benefits from large-scale supervised training and achieves more accurate rotation and translation under strict thresholds. However, our fully unsupervised approach attains higher translation accuracy under relaxed thresholds (RTA@15) and produces lower errors in intrinsic estimation. For NVS, MVSplat struggles to effectively fuse information from 6 context views, whereas our model naturally handles variable-length video inputs and delivers superior rendering results.

Table 5: Quantitative comparison under six-frame input on RealEstate10K (Zhou et al., 2018).

| Method | RRA@5↑ | RRA@15↑ | RTA@5↑ | RTA@15↑ | AbsRel fx↓ | AbsRel fy↓ | Method | PSNR↑ | SSIM↑ | LPIPS↓ |
|---|---|---|---|---|---|---|---|---|---|---|
| DUSt3R | **93.9** | **97.6** | **47.1** | 70.8 | 0.204 | 0.206 | MVSplat | 20.65 | **0.769** | 0.236 |
| Ours | 80.7 | 94.4 | 39.8 | **78.9** | **0.178** | **0.156** | Ours | **23.47** | 0.744 | **0.171** |

To evaluate the generalizability of our method and the accuracy of the learned depth, we perform the zero-shot depth estimation on the KITTI test set with the available ground truth depth, as shown in Tab. 6. The models are trained solely on RealEstate10K (Zhou et al., 2018), making this a cross-dataset evaluation. The results demonstrate that our network learns accurate depth and achieves strong cross-dataset performance on KITTI.

Table 6: Zero-shot depth estimation on KITTI (Geiger et al., 2012).

| Method | abs_rel↓ | sq_rel↓ | RMSE↓ | RMSE log↓ | $\delta < 1.25$ ↑ | $\delta < 1.25^2$ ↑ |
|---|---|---|---|---|---|---|
| SelfSplat | 0.501 | 15.84 | 16.77 | 0.504 | 47.6 | 73.9 |
| Ours | **0.136** | **1.112** | **6.753** | **0.220** | **81.3** | **93.5** |

To further evaluate the cross-dataset performance, we also test the model trained on RealEstate10K (Zhou et al., 2018) on the ACID (Liu et al., 2021) dataset, as shown in Tab. 7. We select two context views with a temporal gap of 80 frames and use the middle frame as the target view to perform target-aligned evaluation. The results show that our model achieves better rendering quality and more accurate camera rotation estimation, while the camera translation estimation is less accurate. This suggests that the camera trajectories of the two datasets may differ significantly, and that our model trained with multi-frame video sequences rather than image pairs is more sensitive to such trajectory discrepancies.

Table 7: Cross-dataset evaluation on ACID (Liu et al., 2021).

| Method | PSNR↑ | SSIM↑ | LPIPS↓ | RRA@5↑ | RRA@15↑ | RTA@5↑ | RTA@15↑ |
|---|---|---|---|---|---|---|---|
| SelfSplat | 23.65 | 0.691 | 0.277 | 79.8 | 92.7 | **42.6** | **68.6** |
| Ours | **24.94** | **0.717** | **0.197** | **80.7** | **94.2** | 22.2 | 59.5 |

## B.2 ADDITIONAL ABLATION STUDY

To investigate the effect of potential information leakage between context views and target views, we conduct an ablation study by replacing the per-frame transformer with a transformer operating over all input frames. The results are reported in Tab. 8. We observe that such information leakage still limits the network's ability to learn accurate 3D representations.

Table 8: Ablation study on potential information leakage between context views and target views.

| Network | PSNR↑ | SSIM↑ | LPIPS↓ | RRA@5↑ | RTA@5↑ |
|---|---|---|---|---|---|
| All-frame Transformer | 21.06 | 0.677 | 0.221 | 65.2 | 25.0 |
| Per-frame Transformer | **23.96** | **0.778** | **0.145** | **85.5** | **50.3** |

We also conduct a cross-dataset ablation study. Specifically, we evaluate the model trained on the RealEstate10K (Zhou et al., 2018) dataset and test it on DL3DV (Ling et al., 2024), as shown in Tab. 9. Since the w/o Stage 2 variant only predicts latent cameras, we evaluate the rendering quality under the target-aware setting. The results show that the model with Stage 2 achieves better rendering performance. This may be because the domain gap between the two datasets introduces errors in the predicted latent space, while Stage 2 aligns the cameras to a unified physical coordinate system, allowing the network to model both datasets more consistently.

Table 9: Ablation study on cross-dataset evaluation. Models are trained on RealEstate10K (Zhou et al., 2018) and tested on DL3DV Ling et al. (2024).

| Network | PSNR↑ | SSIM↑ | LPIPS↓ |
|---|---|---|---|
| w/o Stage 2 | 21.18 | 0.620 | 0.206 |
| w/ Stage 2 | **22.23** | **0.672** | **0.182** |

To demonstrate the scalability of our model, we train it on the combined dataset of RealEstate10K (Zhou et al., 2018), DL3DV Ling et al. (2024), and KITTI (Geiger et al., 2012), and perform target-aware evaluation on RealEstate10K (Zhou et al., 2018), as shown in Tab. 10. These three datasets exhibit significant domain gaps, and KITTI in particular contains substantial interference from moving objects. The performance improvement confirms that our network is able to scale effectively with more diverse data.

Table 10: Ablation on training data size.

| Training Datasets | PSNR↑ | SSIM↑ | LPIPS↓ |
|---|---|---|---|
| RealEstate10K | 26.62 | 0.846 | 0.113 |
| RealEstate10K + DL3DV + KITTI | **27.02** | **0.856** | **0.108** |

## B.3 IMPLICIT CORRESPONDENCE

We visualize the gradients to examine the implicit correspondences learned by our View Synthesis Transformer after Stage 1 pretraining. Specifically, we backpropagate a gradient of 1 from a selected pixel in the predicted image to the shortcut RGB channels of the transformer's input map. We then identify the pixel with the highest absolute gradient as the corresponding point, as illustrated in Fig. 5. The results indicate that the transformer implicitly learns approximate correspondences.

However, as shown in Tab. 11, our further quantitative results show that the learned implicit correspondences are not sufficiently accurate. We randomly sample 100 pairs of implicit correspondences and use RANSAC with the eight-point algorithm to recover the camera extrinsics. For comparison, we also report the pose results directly predicted after Stage 2 alignment. The results show that the implicit correspondences learned in Stage 1 only succeed in recovering camera extrinsics in a limited number of scenes, with performance significantly lower than that of direct pose predictions after

Input Image $\mathbf{I}_1$    Predicted Image    Input Image $\mathbf{I}_2$

Figure 5: Implicit Correspondence. The pretrained View Synthesis Transformer implicitly learns approximate correspondences.

Table 11: Evaluation of implicit correspondence. Camera extrinsics can be recovered in a few cases using implicit correspondences, but there remains a significant gap compared to the predicted pose after Stage 2 alignment.

| Method | RRA@5↑ | RRA@15↑ | RTA@5↑ | RTA@15↑ |
|---|---|---|---|---|
| Stage 1 + RANSAC + 8-point | 40.7 | 86.4 | 8.60 | 34.6 |
| Stage 1 + Stage 2 Prediction | 78.5 | 96.2 | 41.0 | 79.6 |

Stage 2 alignment. This is primarily because the patch-based transformer lacks pixel-level matching capability, and the view synthesis model does not enforce explicit 3D consistency.

## C    ADDITIONAL IMPLEMENTATION DETAILS

We now provide a detailed description of our implementation for readers to reproduce it efficiently. All layers are implemented without bias terms.

### C.1    TRANSFORMER

All of our transformers adopt an architecture similar to GS-LRM, with some modifications. The input maps are transformed into output maps of the same resolution. Specifically, each input map is divided into $8 \times 8$ patches. Each patch is flattened and passed through a linear layer followed by layer normalization (LN) to produce tokens in the hidden dimension. These tokens are then processed by a stack of transformer blocks.

While the hidden dimensions and number of block layers vary across different transformers, the block structure remains consistent. Each block consists of a multi-head attention (MHA) module and a multi-layer perceptron (MLP). The MHA uses 16 attention heads and applies pre-RMS normalization to the keys and queries. The MLP has two layers: the first projects the token to a 4096-dimensional space, followed by a GELU activation, and the second projects it back to the original hidden dimension. Both the MHA and MLP are preceded by the LN layer, and each is followed by a residual connection that adds the module's output back to its input.

Finally, the tokens are unpatchified into output maps. Before unpatchifying, we apply a layer normalization (LN) followed by a linear layer that projects each token to the desired dimension.

Regarding the hidden dimensions and number of layers, the Per-frame Transformer consists of 12 blocks with a hidden dimension of 768. The Camera Transformer, Context Transformer, and View Synthesis Transformer each consist of 8 blocks with a hidden dimension of 512. The context features have the same spatial resolution as the input images, with a channel dimension of 16.

**Shortcut.** All transformers include a shortcut connection from the middle-layer tokens to the final output tokens. Specifically, the intermediate tokens are concatenated with the final-layer tokens and fused via a linear layer. The Per-frame Transformer, Camera Transformer, and Context Transformer each concatenate the corresponding video frame to the output maps to preserve color information.

## C.2 CAMERA DECODER

The Camera Decoder consists of two linear layers and a global average pooling (GAP) operation. The first linear layer projects the input feature map to a 512-dimensional space, which is then aggregated into a vector via GAP. The second linear layer maps this vector to 9-dimensional raw camera parameters: a 4D quaternion $\tilde{\mathbf{R}}^q$ for rotation, a 3D vector $\tilde{\mathbf{t}}$ for translation, and two focal lengths $\tilde{f}_x$ and $\tilde{f}_y$. The weights of the final linear layer are initialized to zero, except for the first element corresponding to the rotation, which is initialized to 1.

The final camera parameters are obtained by applying the following activation functions:

$$\mathbf{R}^q = \frac{\tilde{\mathbf{R}}^q}{||\tilde{\mathbf{R}}^q||_2} \tag{12}$$

$$\mathbf{t} = \tilde{\mathbf{t}} \tag{13}$$

$$f_x = W \times (2.9 \times \mathrm{sigmoid}(\tilde{f}_x) + 0.1) \tag{14}$$

$$f_y = H \times (2.9 \times \mathrm{sigmoid}(\tilde{f}_y) + 0.1), \tag{15}$$

where $W$ and $H$ denote the width and height of the video.

## C.3 GS DECODER

For each context feature map, we predict pixel-aligned Gaussian primitives using a CNN of three layers. Each layer consists of a $3 \times 3$ convolution, an Instance Normalization (IN), and a GELU activation, except for the final layer, which omits the GELU. All intermediate layers use a hidden dimension of 512. Finally, an additional $3 \times 3$ convolution maps the features to the 11-dimensional raw outputs, which include the opacity $\tilde{\alpha} \in \mathbb{R}$, rotation quaternion $\tilde{\mathbf{q}} \in \mathbb{R}^4$, depth $\tilde{D} \in \mathbb{R}$, scale $\tilde{\mathbf{s}} \in \mathbb{R}^2$, and spherical harmonics (SH) coefficients $\tilde{\mathbf{c}} \in \mathbb{R}^3$.

The final Gaussian primitive parameters are obtained by:

$$\alpha = \mathrm{sigmoid}(\tilde{\alpha}) \tag{16}$$

$$\mathbf{q} = \frac{\tilde{\mathbf{q}}}{||\tilde{\mathbf{q}}||_2} \tag{17}$$

$$D = \frac{1}{0.99 \times \mathrm{sigmoid}(\tilde{D}) + 0.01} \tag{18}$$

$$\mathbf{s} = (14.5 \times \mathrm{sigmoid}(\tilde{\mathbf{s}}) + 0.5) \odot \begin{bmatrix} D/W \\ D/H \end{bmatrix} \tag{19}$$

$$\mathbf{c} = \tilde{\mathbf{c}} + \mathrm{RGB2SH}(\mathbf{I}), \tag{20}$$

where $\mathbf{D}$ is used to project the center point according to Eq. (9), and $\odot$ denotes element-wise multiplication. RGB2SH converts RGB values to zeroth-order spherical harmonics. Note that we slightly abuse the notation by using $\mathbf{I}$ to represent the RGB values of the corresponding pixel.

## C.4 TRAINING DETAILS

We implement our network using PyTorch and optimize it with Adam using $\beta_1 = 0.9$ and $\beta_2 = 0.95$. The LPIPS loss is weighted by 0.5, the depth smoothness loss by 0.001, and the projection loss starts with a weight of 1, which linearly decays to 0 after 75% of the training iterations. All models are trained on 16 NVIDIA H800 GPUs, with a batch size of 2 per GPU, resulting in a total batch size of 32. Stage 1 takes 1 day to converge, while Stage 2 requires 1.5 days. We apply gradient accumulation every 4 iterations. Each training stage consists of 200,000 iterations. The learning rate follows a cosine annealing schedule, starting from a peak of $4 \times 10^{-4}$ and decaying to $1 \times 10^{-5}$. A warm-up phase of 2,000 iterations is applied only at the beginning of Stage 1 pertaining.

In fact, as shown in Tab. 12, the pretraining stage can converge even further given a longer number of iterations.

Table 12: Effect of pretraining iterations under target-aware evaluation.

| Iterations | PSNR↑ | SSIM↑ | LPIPS↓ |
|---|---|---|---|
| 200,000 | 25.72 | 0.815 | 0.127 |
| 400,000 | 27.30 | 0.858 | 0.107 |

## D  LIMITATIONS

We assume that input videos depict static scenes, and thus our method is not applicable to the reconstruction of dynamic environments, which remains an interesting future direction.

In addition, our network is deterministic and focuses solely on the reconstruction task without any generative capability. As a result, large view motions will lead to blurry predictions in the unseen regions that are not visible in the context views, as shown in Fig. 6.

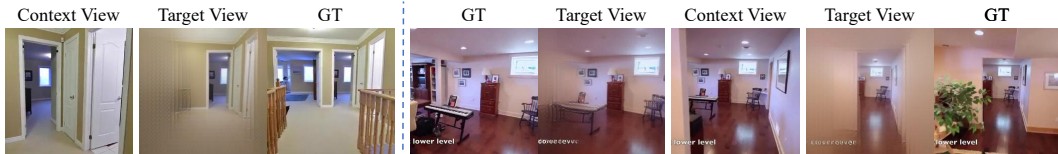

Figure 6: When the target view exhibits significant motion relative to the context views, regions that are unseen in the input become poorly constrained, leading to blurry or ambiguous reconstructions.

## E  LLM USAGE

LLMs were employed solely for grammar checking and minor text polishing, and all LLM-edited content was further reviewed and revised by humans.

