# OpenReview forum: "Pensieve: Self-supervised Novel View Synthesis via Implicit and Explicit Reconstruction"
_ICLR.cc/2026/Conference — Submitted to ICLR 2026_

### Official Review · Reviewer_VEB2 · 2025-10-30

**Soundness:** 3
**Presentation:** 3
**Contribution:** 3
**Rating:** 4
**Confidence:** 4

**Summary:**

The paper presents a solution for novel view-synthesis (NVS) from raw, uncalibrated videos without using any additional camera parameters or geometric priors. The proposed solution consists of two stages. First, the model predicts per-frame latents for the camera and the scene context. Then an implicit renderer (LVSM) is used as an implicit renderer. This first stage provides a strong initialization that helps overcome common optimization challenges. The second stage is a per-frame, per-pixel predictor of 3D Gaussian parameters. By adding this second stage the model aligns the latent space with the real-world through an explicit representation, enforcing 3d consistency. The experiments show competitive quality in camera pose estimation and state-of-the-art results on NVS.

**Strengths:**

1. The paper addresses a very challenging and hot topic in 3D vision: fully self-supervised learning from uncalibrated videos. The core idea of "implicit pretraining, explicit alignment"  is a novel and clever strategy to tackle a known bottleneck. It sidesteps the poor-gradients/local-minima issues of purely explicit self-supervised methods by using an end-to-end implicit renderer (LVSM-like) first to learn a good latent space.

2. The paper's central claims about its own design are exceptionally well-supported by ablations. The "w/o Stage 1" and "w/o Stage 2" experiments in Tables 3 and 4 compellingly demonstrate that both stages are essential and complementary.

3. The method achieves state-of-the-art results (Tables 1 & 2)  against competing methods. Crucially, it does so without the priors that competitors require, such as camera intrinsics (K), pretrained depth (D) and matching (M) networks, or camera poses (P).

4. The paper is well written overall and clearly embedded in the existing literature.

**Weaknesses:**

1. The evaluations seem misleading to me. The paper's main results ("Ours*") in Tables 1 and 2 are achieved by freezing the network and running a 40-iteration, per-scene optimization to refine the test-view camera pose. This is a form of test-time optimization that is not feed-forward. This makes the comparison to feed-forward methods, such as PF3plat, misleading. The true, feed-forward "Ours" result on RealEstate10K (22.20 PSNR) is significantly lower than the optimized "Ours*" (23.96 PSNR) and is actually worse than the competitor PF3plat (22.84 PSNR). This needs to be made much clearer in the results tables and abstract.

2. The inference time solution for two images seems hacky to me. It involves running the network twice—first on 2 frames, then using the LVSM branch to render a new middle frame, and finally re-running the network on all 3 frames. The ablation in Table 3 shows this trick provides a large boost (23.25 to 23.96 PSNR), meaning it is critical for the reported "Ours*" scores. This is computationally expensive and less elegant than a single-pass solution.

3. I am a bit confused about why using the implicit model as the default for the evaluation is supposed to be better. The paper states that "all our NVS evaluations are rendered using the implicit branch". However, in Table 1, the feed-forward explicit branch ("Ours-explicit") achieves a 23.11 PSNR, which is better than the feed-forward implicit branch ("Ours") at 22.20 PSNR. This makes the default use of the implicit branch for evaluation confusing.

4. It would be helpful to include a description of the baselines and how they were retrained (if at all) on this new task.

5. The paper's experimental validation feels incomplete when compared to the baselines it cites. For instance:
The SelfSplat paper also provides strong results on the ACID dataset, reporting an average PSNR of 26.71. The Pensieve paper does not include this dataset, which limits the comprehensiveness of the comparison. More importantly, the SelfSplat paper includes a cross-dataset generalization benchmark (e.g., training on RE10k and testing on ACID), achieving 26.60 PSNR. This is a much stronger test of a model's generalization. Pensieve lacks this critical evaluation, making it difficult to assess whether its two-stage strategy truly generalizes or overfits to the RE10k data distribution.

**Questions:**

1. My primary concern is the evaluation. I would like to see a more honest evaluation in the form of a table that directly compares the purely feed-forward "Ours" model (i.e., no 40-step pose optimization and no IF trick) against the feed-forward results of PF3plat and SelfSplat. This would be a much fairer "apples-to-apples" comparison of the generalizable models.

2. Following up on the question above, how much of the final "Ours*" performance gain comes from the novel two-stage training strategy versus the 40-step inference-time pose optimization? The large gap between "Ours" (22.20) and "Ours*" (23.96) in Table 1 suggests the optimization is a major contributor.

3. What is the computational overhead (e.g., in inference time or FLOPs) of the "Interpolated Frame Enhanced Prediction" (IF) trick?

4. Why is the implicit (LVSM) branch used for the default NVS evaluation when the feed-forward explicit (GS) branch ("Ours-explicit") appears to achieve a better PSNR (23.11 vs 22.20 in Table 1)? What is the performance of the optimized "Ours-explicit*" model?

5. The paper states the "w/o Stage 1" model "fails to converge". This is a key justification for Stage 1. Does this mean the loss diverges, or does it simply converge to a very poor local minimum (as suggested by the 12.97 PSNR in Table 3 )? Could this be due to large viewpoint changes or to the prediction of many redundant Gaussians under the per-pixel GS parameter prediction scheme?

6. Following up on the question above, do we need to predict per-pixel Gaussians? In my opinion, this creates many problems down the line, such as the risk of losing multi-view consistency when dealing with long videos.

7. Could the authors please clarify the discrepancy in the SelfSplat baseline results? This paper reports 22.04 PSNR on RealEstate10K, while the v5 SelfSplat paper reports 24.22 PSNR. Which version of SelfSplat was used for the comparison? Was the SelfSplat baseline retrained from scratch using the Pensieve training curriculum (e.g., unordered frames, N in [2,7])? If so, this would be a fair comparison, but it should be stated explicitly to explain the performance difference. If an older version was used, how do Pensieve's results compare with those from the latest version of SelfSplat?

---

> ### Author Response · Authors · 2025-11-19
>
> Thank you for your review and constructive suggestions. We have revised the paper accordingly and would like to clarify the points you raised as follows:
>
> [W1 & Q1 & Q2] Test-view Optimization (results with \*). We agree that test-time optimization is not a feed-forward process. Therefore, for every result that uses test-time optimization (\*), we report the corresponding results without optimization immediately above it (same name without \*), so readers concerned about this can clearly compare the results.
>
> Regarding the statement that “Ours” (22.20 PSNR) is worse than PF3Splat (22.84 PSNR), this comparison mixes different evaluation protocols. As described in Section 4.2, PF3Splat achieves 22.84 PSNR under the target-aware evaluation setting. Under this same setting, our implicit and explicit branches achieve 26.53 PSNR and 23.21 PSNR, respectively—without using the “Interpolated Frame Enhanced Prediction” (IF) trick or any test-time optimization. The comparison is fully fair and demonstrates that our method performs better in the same evaluation setup.
>
> [W2 & Q3] Interpolated Frame Enhanced Prediction (IF) Trick. The IF trick is only used when estimating camera poses from two-frame inputs, and it adds ~0.034s of inference overhead. Our model relies on this trick for two-frame inputs because our method performs a **single** inference pass on **variable-length inputs** during training. In contrast, baselines like PF3plat and SelfSplat are trained on **fixed 3 frames** and require running the camera network **twice** during training.
>
> If we were to force our network to run twice only for the special case of 2-frame inputs during training, it would create an inconsistency in the training strategy for videos of different lengths and potentially degrade performance on other inputs.
>
> The IF trick provides a simple and efficient way to handle the special two-frame case while keeping our entire training framework elegant and consistent.
>
> [W3 & Q4] Implicit vs. Explicit Branch Performance. There appears to be a misunderstanding regarding the results. In all our experiments, the implicit branch consistently outperforms the explicit branch. In Table 1, Ours-explicit yields 21.36 PSNR, which is worse than Ours(implicit) at 22.20 PSNR. The value of 23.11 PSNR corresponds to Ours-explicit\*, while the corresponding Ours\* (implicit with optimization) achieves 23.96.
>
> [W4] Baseline Weights and Training. For all standard evaluation tasks, we utilized official code and pre-trained weights without retraining, as the evaluation tasks matched their official training settings. The only exception is the DL3DV evaluation. Since PF3plat and SelfSplat were not originally fine-tuned on the full DL3DV dataset, we fine-tuned them on the complete dataset to ensure a fair comparison against our method.
>
> [W5] On completeness of cross-dataset experiments. We highlight that Table 6 already presents cross-dataset experiments, where we evaluate depth estimation accuracy in this setting for the first time. Following your suggestion, we additionally include cross-dataset experiments where the model is trained on RE10K and tested on ACID in Table 7. The results show that our method achieves superior rendering quality and camera rotation accuracy, while translation estimation is less accurate. This is likely due to differences in camera trajectories across datasets, and our model, which is trained on multi-frame sequences rather than image pairs, is naturally more sensitive to shifts in trajectory distributions.
>
> [Q5] Clarification on "Fails to Converge". “Fails to converge” means that the model converges to a poor local minimum, as shown in Table 3 and Figure 3. We agree with your intuition regarding the causes: (1) large viewpoint changes make finding correspondence difficult, and (2) excessive redundant Gaussians hinder gradient propagation.
>
> [Q6] Per-pixel Gaussians. We agree that predicting per-pixel Gaussians is unnecessary. We view the problem of finding an elegant feed-forward mechanism to predict and merge Gaussians as a meaningful but independent research direction. While we have not focused on this in the current work, we regard it as a promising future work.
>
> [Q7] Evaluation differences for RealEstate10K. In the RealEstate10K evaluation, we used the official weights for SelfSplat and did not retrain it. The performance difference arises because their test split is different from the widely used split in PixelSplat, MVSplat, GSLRM, etc. Specifically, SelfSplat fixes a 40-frame interval between the two context views, as stated in their Supplementary A.4.
>
> Our evaluation does not involve variable-length inputs and unordered frames, since SelfSplat and PF3Splat do not support these settings. In contrast, our method supports them naturally and is not specially tuned for the test task.
>
> Thank you again for your time and detailed feedback. We hope our responses address your concerns, and we welcome any further discussion if needed.

---

> ### Author Response · Authors · 2025-11-26
>
> Dear Reviewer VEB2,
>
> Thank you once again for your thoughtful review. As we are now in the final week of the discussion period, we would like to kindly check whether our rebuttal has addressed your concerns. If any remaining questions need further clarification, we would be very glad to continue the discussion.
>
> We sincerely appreciate the time and effort you have devoted to reviewing the paper.

---

> > ### Comment · Reviewer_VEB2 · 2025-11-26
> > **Thank you for the rebuttal**
> >
> > I appreciate the authors' extensive efforts and added experiments, which have successfully addressed many concerns, particularly those regarding cross-dataset generalization and the importance of Stage 1 (implicit pre-training).However, I am not yet fully convinced that the core technical contribution is sufficiently demonstrated, and I have a few remaining questions that need clarification before I can raise my score.
> >
> > 1. Quantification of Test-Time Optimization: The most significant ambiguity lies in the reliance on Test-Time Optimization. The jump from the feed-forward  (22.20 PSNR) to the optimized (23.96 PSNR) is substantial. This obscures the true, generalizable performance of the proposed two-stage training strategy alone. Can the authors quantify, as a percentage, how much of the performance gain is attributable solely to the 40-step test-time optimization? This is critical for assessing the actual contribution of the feed-forward network.
> > 2. The comparison with SOTA methods remains unclear due to differing test splits (SelfSplat) and versioning. To ensure a fair comparison against the current state-of-the-art, could the authors please provide a comparison against the latest, strongest version of SelfSplat? This could involve retraining SelfSplat under the proposed method's full variable-length training curriculum, or clearly citing why the reported 22.04 PSNR is appropriate compared to higher published numbers.

---

> > > ### Author Response · Authors · 2025-11-27
> > >
> > > We thank Reviewer VEB2 for the active engagement in the discussion. We would like to provide the following clarifications regarding the new questions raised.
> > >
> > > 1. Test-Time Optimization (TTO). Below is a summary of the results from all tables regarding the usage of TTO:
> > > | Table | Evaluation     | Method         | TTO Used | PSNR Improvement      |
> > > | ----- | -------------- | -------------- | -------- | --------------------- |
> > > | Table 1     | Target-aware   | Ours-explicit  | No       | 23.21 (Unchanged)     |
> > > | Table 1     | Target-aware   | Ours           | No       | 26.53 (Unchanged)     |
> > > | Table 1     | Target-aligned | Ours-explicit  | No       | 21.36 (Unchanged)     |
> > > | Table 1     | Target-aligned | Ours-explicit* | Yes      | 21.36 → 23.11 (+8.2%) |
> > > | Table 1     | Target-aligned | Ours           | No       | 22.20 (Unchanged)     |
> > > | Table 1     | Target-aligned | Ours*          | Yes      | 22.20 → 23.96 (+7.9%) |
> > > | Table 2     | Target-aware   | Ours           | No       | 21.77 (Unchanged)     |
> > > | Table 2     | Target-aligned | Ours           | No       | 19.36 (Unchanged)     |
> > > | Table 2     | Target-aligned | Ours*          | Yes      | 19.36 → 20.02 (+3.4%) |
> > >
> > > No TTO was used in any other tables (Tables 3-12).
> > >
> > > In summary, TTO is applied only in the Target-aligned Evaluation and is optional. Results using TTO are marked with an asterisk (*), while we explicitly provide the non-TTO results in the immediately preceding row. Readers concerned about the impact of TTO can identify the true, generalizable performance of the proposed method in the row immediately preceding the TTO results.
> > >
> > > 2. Comparison with SelfSplat. We appreciate Reviewer VEB2 for pointing out the updates to the SelfSplat code and RE10K weights. We have updated the results in Table 1 using their latest weights. Specifically, the PSNR metrics are updated as follows:
> > >
> > > - Target-aware Evaluation: SelfSplat: 22.04 → 22.54 (Ours: 26.53)
> > >
> > > - Target-aligned Evaluation: SelfSplat: 18.50 → 19.28 (Ours: 22.20)
> > >
> > > The discrepancy with the results reported in the SelfSplat paper stems from their enforcement of a context view interval of 40. This results in greater image overlap, making the test setting easier. When evaluating our method under this specific setting, we obtain the following results:
> > > | Method    | PSNR  | SSIM  | LPIPS |
> > > | --------- | ----- | ----- | ----- |
> > > | SelfSplat | 24.98 | 0.829 | 0.178 |
> > > | Ours      | 26.93 | 0.860 | 0.107 |
> > >
> > > While our reproduction yields results for SelfSplat that are superior to their paper, our method still outperforms it under this setting.
> > >
> > > We once again thank Reviewer VEB2 for the thorough comments and questions. We hope this response addresses the concerns, and we welcome any further discussion.

---

### Official Review · Reviewer_7i8K · 2025-10-31

**Soundness:** 3
**Presentation:** 3
**Contribution:** 3
**Rating:** 8
**Confidence:** 4

**Summary:**

This paper introduces a novel two-stage training strategy for self-supervised novel view synthesis (NVS), i.e., synthesizing novel views from uncalibrated videos or multi-view unposed images. The key challenge lies in ensuring meaningful 3D scene structure with camera parameters without any explicit supervision, i.e., learning 3D consistency in an implicit reconstruction framework (L.057-L.061). By applying a two-stage training strategy, where an explicit 3D representation (3DGS) is incorporated in the loss functions, the second stage enforces 3D consistency, grounding the model in real-world geometry.

The paper's main contributions are the proposal of this two-stage implicit-to-explicit framework, which achieves state-of-the-art results in high-quality NVS, competitive camera pose estimation, and even the capability of depth estimation, while relying solely on raw video frames for training.

**Strengths:**

The paper is well-written and well-motivated. The core observation—that an implicit pretraining stage can circumvent optimization challenges and provide a robust initialization for a subsequent explicit 3D alignment stage—is a crucial challenge in recent advances of self-supervised novel view synthesis [1,2,3,4]. The experimental setup, including benchmarks (RealEstate10K, DL3DV-10K) and evaluation protocols (Target-aware, Target-aligned), is thorough. The results provide strong evidence for the necessity and effectiveness of the two proposed stages.

[1] LVSM: A Large View Synthesis Model with Minimal 3D Inductive Bias

[2] The Less You Depend, The More You Learn: Synthesizing Novel Views from Sparse, Unposed Images without Any 3D Knowledge

[3] RayZer: A Self-supervised Large View Synthesis Model

[4] No Pose at All: Self-Supervised Pose-Free 3D Gaussian Splatting from Sparse Views

**Weaknesses:**

### 1. Clarity

The paper is well-written, but the clarity could be enhanced by more consistent use of notation. The descriptions in Section 3.3 (L.215-L.242), for example, are purely textual and could be difficult to parse. Incorporating the mathematical notations established in Sections 3.1 and 3.2 would make the methodology easier to follow. Similarly, using notations in the evaluation sections could improve readability; for instance, on L.409, explicitly mentioning that the results of the explicit branch correspond to the rendered image $\hat{I}^G$ would better connect the results to the method. As a minor stylistic point, replacing `\cite` with `\citep` where grammatically appropriate could improve rendering in some browsers.

### 2. LVSM Details

The relationship between the View Synthesis Transformer and the standard LVSM model could be explained more clearly. From Figure 1, it seems the LVSM architecture has been adapted to accept feature maps $F^C$ as input, rather than the raw pixels used in the original version. This suggests the model is not a pretrained, off-the-shelf decoder but is instead an adapted architecture trained from scratch during Stage 1. I believe clarifying these details, either in the main paper or the appendix, would benefit the reader's understanding and aid in reproducibility.

### 3. (Optional) Validation

The experimental results would be more robust with the inclusion of a cross-dataset evaluation. For instance, testing the performance of the model trained on RealEstate10K on the DL3DV dataset would be a strong validation of the method's robustness, e.g., if the camera motion distributions differ significantly between the two datasets, the model with Stage 2 alignment might exhibit superior generalizability compared to the model without Stage 2, even when using a target-aware evaluation protocol.

### 4. Related/Concurrent Work

Discussing some related/concurrent work [1,2] in the main paper or the Appendix could provide a more comprehensive understanding for future research.

[1] RayZer: A Self-supervised Large View Synthesis Model

[2] No Pose at All: Self-Supervised Pose-Free 3D Gaussian Splatting from Sparse Views

**Questions:**

1. There appears to be a slight inconsistency in the reported results for the main model under the Target-aware Evaluation setting. In Table 1, the "Ours" method is listed with a PSNR of 26.53, while in the ablation studies in Table 4, the equivalent "Full" model reports a PSNR of 26.62. Does this reflect a subtle variation in the experimental setup between the main results and the ablation studies, or is it a potential typo?

---

> ### Author Response · Authors · 2025-11-19
>
> Thank you for your review, recognition of our work, and helpful suggestions. We have revised the paper accordingly, and would like to address your concerns as follows:
>
> [W1] On writing clarity. We fully agree with your assessment regarding the clarity of the writing. Following your suggestions, we have rewritten Section 3.3, added necessary notations for better readability, and corrected the citation formatting throughout the paper.
>
> [W2] Phrasing regarding LVSM. Thank you for pointing out the misleading phrasing. We have clarified in the revised Section 3.3 that our LVSM-based branch is trained entirely from scratch. In addition, we have provided the missing details regarding the dimensions of the context features in Section C.1.
>
> [W3] Cross-Dataset Evaluation. Following your recommendation, we conducted experiments where the model is trained on RE10K and tested on DL3DV, comparing the w/ and w/o Stage 2 variants. The results, added to Section B.2, confirm your expectation that the model with Stage 2 achieves superior performance.
>
> We also conducted a cross-dataset evaluation on the ACID dataset (Section B.1). The results show that our method achieves better rendering quality and more accurate camera rotation estimation, while translation estimation is less accurate. This is likely due to differences in camera trajectories between the datasets, and our model, which is trained on multi-frame sequences rather than image pairs, is naturally more sensitive to such trajectory distribution shifts.
>
> Additionally, we highlight that the depth estimation experiments in Table 6 are also conducted under a cross-dataset setting, demonstrating robust generalization capabilities of our method.
>
> [W4] Concurrent Works. Thank you for pointing out the closely related concurrent works. We have included them in Section 2 of the revised paper.
>
> [Q1] Discrepancy between Table 1 and Table 4. We apologize for the omission of this detail in the text. The slight discrepancy arises from the different input configurations used for fair comparison vs. ablation. In Table 1 (target-aware evaluation), the baselines (SelfSplat and PF3plat) only support a triplet input of the form context_1 – target – context_2. However, the standard RE10K test split provided in PixelSplat/MVSplat contains three target views between two context views. To ensure a fair comparison in Table 1, we restricted our method to this triplet setting, running the inference three separate times (once for each target view).
>
> In contrast, since our method natively supports variable-length inputs (context_1 - target_1, target_2, target_3 - context_2), we utilized this more efficient sequence input form for the ablation studies in Table 4.
>
> Thank you again for your time and valuable insights. We hope these responses have addressed your questions, and we are happy to discuss further if needed.

---

> > ### Comment · Reviewer_7i8K · 2025-11-21
> >
> > I appreciate the authors’ efforts during the rebuttal. My concerns have been well addressed, and I believe the work, in its current form, is quite solid. In my view, the most significant contribution lies in grounding a purely implicit view synthesis Transformer in real-world geometry through an insightful—albeit somewhat complex—alignment framework. From the perspective of self-supervised implicit NVS methods (from RUST [1] to Rayzer [2]), where the learned latent pose geometry is interpretable, this work offers a valuable step toward bridging such latent poses with the real-world 3D. To this point, recent frontier work has begun to explore related directions [3], yet from a different perspective. As a result, I believe this paper can meaningfully benefit the community. Despite the strong value, I partially agree with the other reviewers that the extensive technical detail sometimes obscures the central contribution, making it difficult for readers to clearly grasp the paper’s significance. While I understand that self-supervised NVS approaches inherently involve massive details (e.g., complicated evaluation protocols), the paper—albeit detailed and concrete—could be made more concise and focused.
> >
> > Again, I appreciate the authors’ efforts, and I have no further concerns at this time.
> >
> > > [1] RUST: Latent Neural Scene Representations from Unposed Imagery
> > >
> > > [2] RayZer: A Self-supervised Large View Synthesis Model
> > >
> > > [3] True Self-Supervised Novel View Synthesis is Transferable

---

> > > ### Author Response · Authors · 2025-11-26
> > >
> > > Dear Reviewer 7i8K,
> > >
> > > Thank you very much for your thoughtful feedback and positive assessment. We truly appreciate your recognition of our contribution in connecting implicit view-synthesis models with real-world geometry, and we are grateful for the insightful contextualization you provided with respect to recent self-supervised NVS research.
> > >
> > > We fully agree with your suggestion regarding readability, and we will further refine the exposition and figures to make the core ideas clearer and more accessible to readers.
> > >
> > > If any additional questions or thoughts arise, we would be very glad to continue the discussion at any time. Thank you again for your valuable review.

---

### Official Review · Reviewer_DT6P · 2025-11-01

**Soundness:** 3
**Presentation:** 2
**Contribution:** 3
**Rating:** 6
**Confidence:** 3

**Summary:**

This paper proposes a two-stage self-supervised framework for NVS and camera pose estimation from uncalibrated video data, consisting of: (1) implicit reconstruction pretraining and (2) explicit reconstruction alignment. In the first stage, the model learns to synthesize views implicitly through a latent camera representation inspired by LVSM, addressing the optimization limitations of explicit reconstruction methods. In the second stage, it is refined using 3DGS and a depth reprojection loss to enforce 3D geometric consistency. An interpolated-frame strategy further mitigates insufficient camera alignment in extreme two-view settings. Experiments show that the method outperforms prior pose-free and weakly supervised approaches in both view synthesis quality and pose estimation accuracy.

**Strengths:**

- The proposed two-stage framework is well-motivated.
  - The implicit pre-training addresses the optimization challenges/instability of explicit 3D methods.
  - The second stage with 3DGS and depth reprojection enforces explicit geometric consistency and aligns the learned latent representation with the physical 3D space.
- The evaluations are thorough
  - Tab1 and 2 show superior performance over baselines on two datasets, i.e., RealEstate10K and DL3DV-10K.
  - The authors ablate the effectiveness of two stages and IF in Tab3 and 4.

**Weaknesses:**

- Although the authors claim that 3DGS adds geometric consistency, the evaluation relies solely on rendering metrics (PSNR, SSIM, LPIPS), which may not fully capture geometric or temporal fidelity. Incorporating geometry-aware consistency metrics would strengthen the claims.
- The multi-stage training pipeline (4 transformers, multiple losses) may be difficult to reproduce and is computationally demanding.
- Fig1 is difficult to interpret, as it contains too many architectural details that make the overall pipeline visually overwhelming.
- The paper doesn’t analyze failure cases (e.g., in scenes with extreme motion).

**Questions:**

Please find my questions in weakness.

---

> ### Author Response · Authors · 2025-11-19
>
> Thank you for your review. We have revised the paper according to your suggestions, and would like to clarify the points you raised as follows:
>
> [W1] Clarification on Evaluation and Consistency. There may be a slight misunderstanding here. Our evaluation strategy follows the standard evaluation protocols of prior NVS works: we render images at given target views and compute standard rendering metrics. Since the Ground Truth (GT) target images are captured from the real world, they inherently possess correct geometric consistency. Therefore, high rendering quality against these GT images serves as a strong proxy for reconstruction consistency. Furthermore, regarding Stage 2, we emphasize that by introducing an explicit representation (3DGS), we inherently enforce triangulation. Unlike purely implicit methods, explicit representations will be rendered geometrically consistent across different viewpoints. Consequently, Stage 2 effectively aligns the implicit latent space with the real physical space.
>
> [W2] Model Complexity. The four transformers are not independent or competing modules. They are different components within a single end-to-end architecture, each serving a specific function. They could also be implemented as a unified network. The loss terms in the image domain are also cooperative, and they share the same numerical range, so no careful weight tuning is required. The only term that requires a separate weight is the depth smoothness loss, whose weighting can be directly adopted from prior work.
>
> During training, our network converges more easily compared to methods based purely on explicit representations (as shown in Fig. 3). Regarding computational cost, the complexity can be adjusted by scaling the model size.
>
> [W3] Figure 1 Revision. We appreciate your suggestion and fully agree that Figure 1 currently contains unnecessary details. We commit to redesigning it to improve clarity for future readers in the final revision of the paper.
>
> [W4] Failure Cases on Extreme Motion. Thank you for pointing this out. We have added an analysis of failure cases involving extreme camera motion in Section D of the Supplementary Material. Since our network is deterministic, it tends to predict blurry results in regions that are unseen or occluded in the context views.
>
> Thank you again for your time and valuable feedback. We hope this response resolves your queries. We are open to further discussion if any questions remain.

---

> ### Author Response · Authors · 2025-11-26
>
> Dear Reviewer DT6P,
>
> Thank you once again for your positive assessment and constructive feedback. As we are now in the final week of the discussion period, we would like to kindly ask whether our rebuttal has addressed your main concerns. If there are any remaining points that you feel would benefit from further clarification, we would be very glad to discuss them in more detail.
>
> We sincerely appreciate the time and effort you have devoted to reviewing our paper and helping us improve its overall quality.

---

### Official Review · Reviewer_P8P3 · 2025-11-03

**Soundness:** 3
**Presentation:** 3
**Contribution:** 3
**Rating:** 4
**Confidence:** 4

**Summary:**

This paper focuses on the field of learning 3D pose estimation and 3D appearance prediction from unposed and uncalibrated video frames. The key contribution lies in an implicit reconstruction for leaning camera poses with LVSM and an explicit reconstruction module that learns Gaussian attributes.

**Strengths:**

1. This paper focuses on a meaningful task that directly learns 3D reconstruction models from unposed images which are easy to collect in the real-world. The scheme provides a great potential in scaling up to large scale data.
2. The framework is simple, which also benefits  data scaling up.
3. The  results are convincing, outperforming SelfSplat.

**Weaknesses:**

1. The key idea of this paper seems to be replacing the matching-based camera pose estimation part of SelfSplat with the LVSM-based solution. The LVSM is also an off-the-shelf method.  More insights are needed for highlighting the differences.

2. More comparison with DBARF and FlowCAM are required.

3. I am curious with the scaling-up capability of the method. The key advance of the self-supervised learning scheme lies in a flexible scaling-up to unposed real-world images. However, the method are only trained with standard datasets where camera pose are easy to obtain, which does not demonstrates its effectiveness. I understand that the revision time are not enough for large-scale training, yet am still interested in the potential.

**Questions:**

1. How long does the method take to converge in current GPU settings?
2. Does the method work for dense view inputs (e.g. 100 images)?

I am willing to raise my score if the authors can convince me with the scaling-up capabilities and differences with SelfSplat.

---

> ### Author Response · Authors · 2025-11-19
>
> We thank the reviewer for the constructive feedback and the time spent reviewing our work. We have revised the paper according to your suggestions and would like to clarify the following points.
>
> [W1] Differences from SelfSplat. Our method adopts a fundamentally different framework, which naturally supports variable-length inputs and introduces an implicit-reconstruction pretraining strategy. In contrast, SelfSplat’s camera estimator and Gaussian decoder operate on only two input frames, making it difficult to handle videos of variable length.
>
> While our implicit branch is inspired by the general idea of LVSM, the network architecture is different and is trained entirely from scratch. Our key insight is that a fully end-to-end implicit reconstruction framework is beneficial for 3D self-supervised learning (as shown in Fig. 3). We further analyze the gap between the implicit latent space and the explicit physical space, and propose an alignment stage to address this issue.
>
> [W2] Additional baselines. Thank you for suggesting more baselines. Unfortunately, We found that there is no official implementation or pre-trained weights available for DBARF on the RealEstate10k or DL3DV dataset. Following your recommendation, we added a comparison with FlowCam and updated Table 1 accordingly.
>
> [W3] On the model’s scale-up capability.
> Due to time constraints, we were unable to collect a new dataset. Instead, we trained our model on the combined datasets (RE10K + DL3DV + KITTI) and evaluated it on RE10K under the target-aware setting. These datasets have significant domain gaps, and KITTI contains substantial moving-object disturbances. We report the corresponding results in Section B.2, and the table is also shown below:
> | Training Datasets | PSNR ↑ | SSIM ↑ | LPIPS ↓ |
> |------|--------|--------|---------|
> | RealEstate10K    | 26.62  | 0.846  | 0.113   |
> | RealEstate10K + DL3DV + KITTI | **27.02**  | **0.856** | **0.108** |
>
> The performance improvement demonstrates the scaling-up capability of our method.
>
> [Q1] Convergence time. Under our current GPU settings, Stage 1 takes 1 day to converge, and Stage 2 takes about 1.5 days. We have added this information to Section C.4.
>
> [Q2] Dense-view inputs. In principle, our method can be applied to dense-view inputs. However, the Transformer backbone has an $O(N^2)$ complexity with respect to the number of input views, making dense inputs difficult to support under current memory and compute limits. We agree that developing efficient feed-forward reconstruction networks for dense views is a vital research direction, and we regard combining our proposed method with such efficient architectures as promising future work.
>
> Thank you again for your time and valuable feedback. We hope our responses address your concerns, and we are open to further discussion if any questions remain.

---

> ### Author Response · Authors · 2025-11-26
>
> Dear Reviewer P8P3,
>
> We would like to thank you once again for your valuable feedback and insightful questions. As we are now in the final week of the discussion period, we kindly ask whether our rebuttal has addressed your main concerns. If there are any remaining points that you feel require further clarification, we would be more than happy to provide additional details.
>
> We sincerely appreciate the time and effort you have devoted to reviewing our paper and helping us improve its clarity and quality.

---

### Author Response · Authors · 2025-12-01
**Summary of Rebuttal Updates and Clarifications**

Dear Area Chair,

We sincerely appreciate the time and effort you have dedicated to the review process. To assist with your final evaluation, we would like to provide a summary of our rebuttal updates.

**1. Response to Reviewers Willing to Raise Scores**

Reviewers P8P3 and VEB2 (initially rating 4) indicated they are willing to raise their scores if specific concerns are resolved. We summarize our responses to these points below:

**Scaling-up Capability (R-P8P3)**: We demonstrated the model's scalability by training on a combination of three datasets from different domains, which yielded a clear performance boost (PSNR: 26.62 → 27.02).

**Network details (R-P8P3 & R-7i8K)**: Reviewer P8P3 was concerned that our method merely replaces SelfSplat’s matching-based estimation [1] with an off-the-shelf LVSM [2] component. Following R-7i8K’s suggestion, we revised the paper to clearly state that our entire network is trained from scratch. Our model is more flexible and naturally supports variable-length inputs. We further highlighted our key insight: a fully self-supervised implicit reconstruction framework benefits 3D self-supervised learning, and we analyze and address the gap between the implicit latent space and the explicit physical space.

**Cross-dataset Evaluation (R-7i8K & R-VEB2)**: We emphasized that our depth estimation experiments (Table 6) are inherently cross-dataset. Additionally, we conducted further comparisons on the ACID dataset (Table 7) and ablation studies on the DL3DV dataset (Table 9) to support the generalization capability of our method.

**Test-Time Optimization (TTO) (R-VEB2)**: The reviewer raised concerns that TTO might obscure the true feed-forward performance. We clarified that TTO is optional and used only for a subset of results. All results using TTO are explicitly marked with an asterisk (*), and the corresponding non-TTO results are provided immediately above to ensure a fair comparison.

**Comparison with latest SelfSplat (R-VEB2)**: We updated our experiments using the latest SelfSplat weights, and the PSNR improved from 22.04 to 22.54 (ours: 26.53). We clarified that the discrepancy with the numbers reported in the SelfSplat paper arises from their use of a simpler test split. Under that split, our method still outperforms SelfSplat (PSNR: SelfSplat: 24.98 vs. Ours: 26.93).

We have provided detailed responses and additional evidence for all major concerns raised by Reviewer P8P3 and Reviewer VEB2, including the points they identified as prerequisites for updating their scores. Their feedback—and that of the other reviewers—has also helped us substantially clarify and improve the paper.

**2. Improvements Based on Constructive Feedback**

The reviewers' feedback significantly helped us clarify and improve the manuscript:

**Clarity (R-7i8K & R-DT6P)**: We rewrote Section 3.3 and improved citation formatting and notation following Reviewer 7i8K’s suggestion. We also commit to simplifying Figure 1 to enhance readability as suggested by R-DT6P.

**Failure Cases (R-DT6P)**: We added a discussion on failure cases in Section D of the Supplementary Material.

**Interpolated Frame Enhanced Prediction (IF) Trick (R-VEB2)**: We clarified that the IF trick is an efficient (~0.034s) and simple method to handle the special two-frame inference case while preserving the consistency and elegance of our training pipeline.

**Additional Revisions (R-P8P3 & R-7i8K)**: As suggested, we added a new baseline comparison (FlowCam [3]), convergence time analysis, and a discussion of concurrent work.

For all other specific questions, we have provided detailed clarifications in our individual responses. We deeply appreciate the reviewers’ efforts and constructive feedback, and we are grateful for the additional work you have taken on to facilitate the review process, given the unusual circumstances. We hope this summary is helpful for your final assessment.

Best regards,

The Authors

> [1] SelfSplat: Pose-Free and 3D Prior-Free Generalizable 3D Gaussian Splatting
> [2] LVSM: A Large View Synthesis Model with Minimal 3D Inductive Bias
> [3] FlowCam: Training Generalizable 3D Radiance Fields without Camera Poses via Pixel-Aligned Scene Flow

---

### Meta-Review · Area_Chair_zhfq · 2025-12-25

**Summary:**

Reviewers generally found the paper well-motivated and technically sound, highlighting the proposed two-stage implicit-to-explicit framework as a meaningful contribution to self-supervised novel view synthesis and pose estimation from uncalibrated videos. Strengths include strong empirical performance, clear ablations validating both stages, and competitive results on standard benchmarks.
The main concerns focused on evaluation fairness and clarity (notably the role of test-time optimization), comparisons with strong baselines (e.g., SelfSplat), scaling and generalization evidence, and method complexity and clarity of presentation.

**Reviewer Concerns:**

**Addressed by the rebuttal:**

Differences from SelfSplat and LVSM clarified.

Scaling and generalization supported via multi-dataset training and cross-dataset evaluations (RE10K, DL3DV, ACID).

Test-time optimization (TTO) clarified as optional, clearly marked, and quantitatively analyzed.

Baseline comparisons updated using latest SelfSplat weights and additional baselines (e.g., FlowCam).

Clarity and reproducibility improved (rewritten sections, added convergence time, failure case analysis).

**Still partially outstanding:**

Reliance on test-time optimization remains a concern for some reviewers regarding the core feed-forward contribution.

Model complexity and evaluation transparency may still limit accessibility.

Generality of gains without TTO remains debated by at least one reviewer.

**Reviewer Scores:**

Reviewer P8P3: Likely improved after clarifications on scaling, baselines, and differences from SelfSplat.

Reviewer DT6P: Likely unchanged.

Reviewer 7i8K: Explicitly satisfied.

Reviewer VEB2: Likely unchanged.

---

### Decision · Program_Chairs · 2026-01-26

Reject